# Astrocyte-Mediated Neuroinflammation in Neurological Conditions

**DOI:** 10.3390/biom14101204

**Published:** 2024-09-25

**Authors:** Yanxiang Zhao, Yingying Huang, Ying Cao, Jing Yang

**Affiliations:** 1State Key Laboratory of Membrane Biology, School of Life Sciences, Peking University, Beijing 100871, China; 2The Affiliated High School, Peking University, Beijing 100080, China; 3Center for Life Sciences, Academy for Advanced Interdisciplinary Studies, Peking University, Beijing 100871, China; 4Peking University Third Hospital Cancer Center, Beijing 100191, China

**Keywords:** astrocytes, central nervous system, neuroinflammation, cytokines, neurological conditions

## Abstract

Astrocytes are one of the key glial types of the central nervous system (CNS), accounting for over 20% of total glial cells in the brain. Extensive evidence has established their indispensable functions in the maintenance of CNS homeostasis, as well as their broad involvement in neurological conditions. In particular, astrocytes can participate in various neuroinflammatory processes, e.g., releasing a repertoire of cytokines and chemokines or specific neurotrophic factors, which result in both beneficial and detrimental effects. It has become increasingly clear that such astrocyte-mediated neuroinflammation, together with its complex crosstalk with other glial cells or immune cells, designates neuronal survival and the functional integrity of neurocircuits, thus critically contributing to disease onset and progression. In this review, we focus on the current knowledge of the neuroinflammatory responses of astrocytes, summarizing their common features in neurological conditions. Moreover, we highlight several vital questions for future research that promise novel insights into diagnostic or therapeutic strategies against those debilitating CNS diseases.

## 1. Introduction

The central nervous system (CNS), unlike peripheral tissues, is functionally separated from the body’s immune system by barrier structures, including the blood–brain barrier (BBB), blood–cerebrospinal fluid barrier, and meninges [1,2,3,4,5,6]. This unique property of the CNS is crucial for ensuring the homeostasis and stability of its internal environment, preventing any disruptive action of peripheral immune cells or their derived factors. As a result, the CNS must rely on glial cells, particularly microglia and astrocytes, for immune surveillance and responses to various neuropathological insults [7,8,9,10]. Indeed, extensive studies have elucidated the complex functions of microglia, bona fide residential immune cells in the CNS, in diverse inflammatory scenarios, e.g., pathogenic infection, neural damage, autoimmunity, and neurodegeneration. Those critical aspects of microglia-mediated neuroinflammation have been comprehensively reviewed in the literature [11,12,13,14,15,16].

Astrocytes are star-shaped cells that share the same origin as neurons. During the early stage of CNS development, precursor cells differentiate from neural progenitor cells and then mature into astrocytes at the late embryonic and early postnatal stages [17,18]. Traditionally, astrocytes can be classified as fibrous, protoplasmic, and radial subtypes based on their distinct morphologies and anatomical locations [19,20]. In addition, astrocytes can be identified by cellular markers, e.g., glial fibrillary acidic protein (GFAP), vimentin, or S100 calcium-binding protein β (S100β) [21]. However, recent studies based on transcriptomics and advanced imaging techniques have revealed the enormous heterogeneity of astrocytes at the structural, functional, and molecular levels [22,23,24,25]. Also, it has become increasingly recognized that astrocytes possess significant plasticity during regional specialization to the extent that they may exhibit neurocircuit-specific properties [26,27,28,29]. Such topics on astrocyte heterogeneity or plasticity have been recently reviewed and can be referred to for more details [22,23,24,25,26,27,28,29]. For the scope of this review, we will primarily focus on the common aspects of astrocyte immune functions in physiological or neurological contexts.

Although astrocytes are more abundantly present than microglia in the CNS, their immune capacities have only become appreciated over the past two decades [30,31,32]. Accumulating evidence has established that astrocytes can release a repertoire of cytokines, chemokines, reactive oxygen species, and other immune factors, thus being an essential source of neuroinflammation with either beneficial or detrimental outcomes. Here, we will briefly describe the physiological functions of resting “homeostatic” astrocytes and then move on to the key features of astrocyte-mediated neuroinflammation under neurological conditions.

## 2. Physiological Functions of Astrocytes

Under a normal physiological condition, astrocytes adopt a resting “homeostatic” state and exert several indispensable roles in supporting neuronal survival and functions (Figure 1):

### 2.1. Maintenance of the BBB

Astrocytes represent a central component of the BBB [2,6,33]. Astrocytes extend the flattened processes that ensheath most of the blood vessels in the brain. Those unique anatomical structures, known as endfeet, are in close contact with endothelial cells, providing structural support for the BBB. Astrocytes can secrete, via their endfeet, several signaling factors, e.g., glial cell line-derived neurotrophic factor (GDNF) and transforming growth factor β (TGFβ), that promote the formation of tight junctions between endothelial cells [34,35,36]. Also, those astrocyte-derived factors facilitate the functional properties of endothelial cells, including a low rate of transcytosis. In addition, astrocytes contribute to the immune barrier function of the BBB by suppressing the transmigration of immune cells across blood vessels. Moreover, astrocytes can regulate the regional volume of blood flow by releasing vasoactive substances, e.g., arachidonic acid or its metabolites, in response to increased neuronal activities [37,38,39]. This process, commonly known as functional hyperemia, enables neurons actively transducing action potentials to receive adequate oxygen, glucose, and other essential nutrients. In this scenario, astrocytes act as the key metabolic regulator of the BBB to ensure the neurophysiological functions of the brain.

### 2.2. Modulation of Synaptic Transmissions

In addition to their endfeet around blood vessels, astrocytes extend perisynaptic processes that have direct contact with neurons and their synapses. Studies have suggested that every synapse in the CNS is enwrapped by at least one astrocyte, and a single astrocyte may have processes projecting to hundreds of synapses [40,41,42]. Such astrocyte–neuron interactions are critical in modulating synaptic transmissions for the proper actions of neurocircuits. Of importance are several transporter proteins for neurotransmitters, e.g., glutamate, glycine, and gamma-aminobutyric acid (GABA), expressed on those perisynaptic processes of astrocytes, which uptake the neurotransmitters released from a presynaptic terminal during action potentials [43,44,45]. This recycling capacity influences the strength and duration of synaptic transmissions and is indispensable for blocking excitotoxicity. Also, astrocytes can supply surrounding neurons with neurotransmitter precursors such as glutamine. In addition, via their perisynaptic processes, astrocytes can release gliotransmitters, e.g., ATP and *D*-serine, that modulate synaptic strength and plasticity [44,46,47]. Moreover, astrocytes help buffer the ion levels within the tissue environment of CNS [48,49]. For instance, astrocytes uptake extracellular K^+^ released during action potentials, which prevents the potential hyperexcitability of neurons.

### 2.3. Trophic Support to Neurons and Oligodendrocytes

Astrocytes provide vital trophic support to neurons. It has long been recognized that astrocytes can produce neurotrophic factors, particularly brain-derived neurotrophic factor (BDNF), that facilitate neuronal survival and functions [50,51]. Moreover, astrocytes respond to increased neuronal activities by taking more glucose from the blood circulation to produce lactate through glycolysis. Lactate is then transferred from astrocytes to neurons via monocarboxylate transporters to meet the high energy demand of neurons, especially during their intense firing of action potentials. This astrocyte–neuron lactate shuttle has been suggested to sustain synaptic transmissions and plasticity [52,53]. In addition, astrocytes release specific signaling molecules that critically influence the survival and functions of oligodendrocytes. For instance, bone morphogenetic proteins (BMPs) derived from astrocytes modulate the maturation of oligodendrocyte precursor cells (OPCs) and support their myelin formation [54,55].

### 2.4. The Glial–Lymphatic System

Astrocytes are not only involved in the BBB but also form a unique fluid system in the CNS, i.e., the glia–lymphatic system (also known as the glymphatic system). Astrocytes express the water channel protein aquaporin-4 (AQP4) on their endfeet, which enables the fluid exchange between the periventricular space of blood vessels and the interstitial fluid of the CNS [56,57,58]. The glymphatic system helps to distribute glucose, lipids, and other nutrients to neurons distant from blood vessels. Also, this astrocytic action represents a critical mechanism of removing metabolic wastes generated within CNS parenchyma back to the blood circulation. Moreover, it has become recently recognized that the glymphatic system has a crucial role in the clearance of key pathological proteins in Alzheimer’s disease or Parkinson’s disease, e.g., amyloid-beta (Aβ), Tau, and α-synuclein [59,60,61,62].

## 3. Astrocyte-Mediated Neuroinflammation

It has long been observed that upon neurological insults, astrocytes often undergo a stereotyped process known as reactive astrogliosis, characterized by morphological changes, cell proliferation, and neuroinflammation (Figure 2).

### 3.1. Reactive Astrocytes

Compared to the resting “homeostatic” state, reactive astrocytes undergo significant morphological changes, i.e., cellular hypertrophy, extension of main processes, and reduction of perisynaptic processes [19,20,63]. Also, reactive astrocytes elevate their expression of specific marker proteins, such as GFAP, the primary component of their intermediate filaments [21,64]. In addition, reactive astrocytes produce a collection of extracellular matrix proteins, including laminin, fibronectin, collagens, and chondroitin sulfate proteoglycans, which together form the glial scar at a site of neurological insult [65,66,67]. It has been suggested that such scar structures help restrict the infiltration and migration of peripheral immune cells, thus establishing a barrier to protect healthy brain tissues from the collateral damage of inflammatory events. On the other hand, astrocyte-derived proteins in the glial scar, e.g., chondroitin sulfate proteoglycans, strongly inhibit the process of axonal regeneration, potentially impeding the restoration of damaged neurocircuits under neurological conditions [68,69,70,71].

### 3.2. Mechanisms of Astrocyte-Mediated Neuroinflammation

Astrocytes are an integral part of CNS innate immunity, capable of producing a repertoire of cytokines, chemokines, and other immune factors [30,31,32,66]. Even in their resting state, astrocytes can secrete granulocyte colony-stimulating factor (G-CSF), granulocyte-macrophage colony-stimulating factor (GM-CSF), and TGFβ. Under neurological insults, reactive astrocytes respond to pathogen-associated molecular patterns (PAMPs) or damage-associated molecular patterns (DAMPs) via toll-like receptors (TLRs) or other pattern recognition receptors (PRRs). Engagement of those specific receptors activates the downstream immune signaling pathways, e.g., transcription factors of the nuclear factor kappa B (NF-κB) family, activating protein-1 (AP-1) family, or interferon regulatory factor (IRF) family. Those key transcription factors then cooperatively enable the expression of inflammatory cytokines, e.g., tumor necrosis factor-alpha (TNFα), interleukin (IL)-1β, and IL-6, and chemokines, e.g., CCL2, CCL3, CCL5, CXCL8, and CXCL10. Also, inflammasomes constituted by NOD-like receptor proteins (NLRs) such as NLRP3 can be induced during such immune responses. In addition, reactive astrocytes can release ATP and reactive oxygen species such as nitric oxide (NO). As discussed below, those astrocyte-derived immune factors potently trigger the alteration of BBB functions, the recruitment of peripheral immune cells, and other inflammatory events.

Notably, astrocyte-mediated neuroinflammation exhibits remarkable heterogeneity, which may either exacerbate neural damage or promote tissue repair. Studies have postulated that reactive astrocytes can be classified into at least two distinct subtypes, i.e., pro-inflammatory and neurotoxic A1 versus immunosuppressive and neuroprotective A2 [72,73,74]. In particular, A1 reactive astrocytes induced by chronic neurological cues, e.g., neurodegenerative diseases, produce excessive inflammatory cytokines and reactive oxygen species, exaggerating neuronal cell death and the disruption of neurocircuits. On the contrary, A2 reactive astrocytes, which are often encountered upon tissue damage, can facilitate the formation of glial scar, the clearance of cellular debris, and the BBB repair. Therefore, astrocytes have a complex role in neuroinflammation, exerting detrimental or beneficial effects depending on the nature and timing of neurological insults.

### 3.3. Astrocyte–Microglia Crosstalk

Importantly, astrocyte-mediated neuroinflammation also relies on their interactions with other cell types in the tissue microenvironment. In particular, astrocytes and microglia can engage in bidirectional crosstalk during inflammatory processes, with each cell type designating the immune functions of the other one through cytokines, chemokines, and other signaling factors (Figure 2). Accumulating evidence has suggested that astrocyte-mediated neuroinflammation can modulate the activation states of microglia, ranging from anti-inflammatory to pro-inflammatory responses. Also, astrocyte-derived inflammatory factors, e.g., CCL2 and ATP, influence microglial motility and phagocytotic capacity [75,76]. At the same time, microglia can release specific signaling molecules, e.g., TNFα and IL-1α, to switch reactive astrocytes from neuroprotective to neurotoxic properties [74,77,78]. Such reciprocal interactions between reactive astrocytes and microglia may amplify neuroinflammation, leading to the excess production of pro-inflammatory cytokines and reactive oxygen species that exacerbate neural damage. Key aspects of astrocyte–microglia crosstalk in neuroinflammation have been comprehensively summarized in recent review articles [79,80,81,82].

## 4. Astrocytes in Neurological Conditions

Given their essential roles in maintaining CNS homeostasis and prominent capacities of inflammatory responses upon activation, it is unsurprising that astrocytes broadly participate in neurological conditions.

### 4.1. Traumatic Neural Injuries

It has been extensively documented that traumatic injuries to the CNS trigger reactive astrogliosis [83,84,85]. Destruction of neurons and other non-neuronal cells causes the release of DAMPs, e.g., DNA, RNA, and intracellular proteins, eliciting the immune activation of astrocytes [30,31,32]. In line with their functional heterogeneity, reactive astrocytes may exert dual roles in tissue repair and long-term remodeling in this neurological scenario. On the one hand, astrocytes produce a large amount of pro-inflammatory cytokines, e.g., TNFα, IL-1β, and IL-6, and reactive oxygen species, leading to further neural damage within and adjacent to an injury site [86,87,88]. For instance, TNFα potently causes the death of neuronal cells through the signaling pathways of apoptosis or necroptosis [89,90]. Also, reactive oxygen species can damage the mitochondrial function of neurons, leading to their dysfunction and demise [91,92]. In addition, those astrocyte-derived immune factors promote microglial activation and, through the BBB, recruit peripheral immune cells, e.g., monocytes/macrophages, neutrophils, and lymphocytes, together amplifying neuroinflammation. Meanwhile, the transition of astrocytes from the resting state to the reactive state reduces their perisynaptic processes, significantly diminishing their support for normal synaptic functions [19,20,63]. For instance, the reduction of glutamate uptake by astrocytes can result in excitotoxicity that exaggerates neuronal cell death during traumatic injuries [93,94]. Moreover, reactive astrocytes secrete extracellular matrix proteins, e.g., tenascins and chondroitin sulfate proteoglycans, that impede axonal regeneration for neural repair [69,95,96].

In contrast to those detrimental effects, astrocytes release neurotrophic factors, e.g., BDNF and GDNF, that facilitate neuronal survival after traumatic injuries [50,51,97]. Also, astrocytes are capable of producing thrombospondins and glypicans, which facilitate synapse formation and the reconstruction of neurocircuits [98,99,100]. In addition, reactive astrocytes can produce vascular endothelial growth factors (VEGFs), stimulating angiogenesis to restore the blood supply to an injury site [50,101]. Moreover, astrocytes form the glial scar to prevent the spread of inflammatory immune cells and their factors, which protects adjacent neural tissue from collateral damage [65,66,67]. Therefore, the intricate balance between the neurotoxic and neuroprotective functions of astrocytes contributes to the extent and recovery of traumatic neural injuries.

### 4.2. Strokes

Hemorrhagic or ischemic strokes disrupt the vascular integrity of an infarct region [102,103]. Notably, such BBB breakage directly results in the infiltration of peripheral immune cells from the blood circulation. At the same time, loss of the blood supply causes the death of neurons and other non-neuronal cells, which release DAMPs to trigger the inflammatory responses of astrocytes [104,105,106]. Similar to the scenario of traumatic neural injuries, reactive astrocytes produce pro-inflammatory cytokines and reactive oxygen species while reducing their trophic support for neurons, together exacerbating neural damage. Importantly, such astrocyte-derived inflammatory factors can impinge on the tight junctions of endothelial cells [107,108,109]. In addition, reactive astrocytes undergo morphological changes to retract their endfeet contacting blood vessels, which further compromises the normal function of the BBB [19,20,107].

On the other hand, astrocyte-mediated neuroinflammation also exerts a crucial role in tissue repair and neuroprotection during strokes. Reactive astrocytes secrete extracellular matrix proteins to form the glial scar, limiting the infiltration of peripheral immune cells due to the BBB breakdown [65,66,67]. Meanwhile, astrocytes express VEGFs that facilitate angiogenesis for the BBB repair after stroke damage [50,101]. In addition, reactive astrocytes can produce anti-inflammatory cytokines, e.g., IL-10, to ameliorate the inflammatory events in an infarct region [110,111]. Moreover, astrocyte-derived neurotrophic factors BDNF and GDNF may promote neuronal survival and the restoration of neurocircuits [50,51,97].

### 4.3. Neurodegenerative Diseases

Astrocyte-mediated neuroinflammation is broadly involved in neurodegeneration, e.g., Alzheimer’s disease, Parkinson’s disease, and amyotrophic lateral sclerosis.

#### 4.3.1. Alzheimer’s Disease (AD)

Reactive astrocytes with the morphology of reduced cellular processes are observed in post-mortem brain tissues of AD patients [112,113,114]. Importantly, resting astrocytes are an indispensable part of the glymphatic system that facilitates the clearance of the hallmark pathological proteins of AD, i.e., Aβ and Tau, and loss of this homeostatic function exaggerates the Aβ and Tau aggregates [59,60,61]. Also, the astrocytic modulation of synaptic functions via neurotransmitter recycling or gliotransmitter signals become compromised, causing excitotoxicity for neuronal loss in the disease progress of AD [115,116]. Indeed, the disease-associated phenotype of astrocytes could be identified by single-nucleus RNA sequencing at early stages of Alzheimer’s disease models [117]. In further support of their involvement in AD, astrocytes are a major source of apolipoprotein E (APOE) in the CNS [118,119]. It has been well documented that human APOE has three alleles, with APOE4 being the strongest genetic risk factor for late-onset AD [119,120,121]. Accordingly, selective blockage of the *ApoE4* expression in astrocytes can delay the deposition of Aβ and Tau aggregates in mouse models [122].

Studies have suggested that Aβ aggregates engage the receptor for advanced glycation end products (RAGE) or TLRs on astrocytes, triggering the inflammatory response via the NF-κB pathway [123,124,125]. Also, neuronal cell death releases DAMPs that can sustain the chronic activation of astrocytes [30,31,32]. Notably, both A1 and A2 astrocytes are present in the disease context of AD [126,127,128]. A1 astrocytes produce pro-inflammatory cytokines and reactive oxygen species that contribute to neural damage. In contrast, A2 astrocytes primarily release anti-inflammatory cytokines and neurotrophic factors, thus exerting a neuroprotective effect. In addition, astrocyte-derived immune factors, e.g., IL-3, may stimulate the microglial ability to clear Aβ and Tau proteins [129,130,131]. Therefore, targeting astrocyte-mediated neuroinflammation to reduce pro-inflammatory factors and oxidative stress while restoring their support for neuronal survival is essential to limit AD pathology.

#### 4.3.2. Parkinson’s Disease (PD)

Reactive astrocytes are prominently present in the inflicted brain regions, e.g., substantia nigra and striatum, of PD patients [132,133]. Similar to the scenario of AD, resting astrocytes act in the glymphatic system to clear the key pathological protein of PD, i.e., α-synuclein, and the disturbance of this critical mechanism exacerbates the α-synuclein accumulation for dopaminergic neuronal death [59,62]. Meanwhile, α-synuclein aggregates or DAMPs derived from neuronal loss trigger the immune activation of astrocytes, leading to their excess production of pro-inflammatory cytokines and reactive oxygen species [134,135,136]. Accordingly, it has been postulated that strategies to enhance α-synuclein clearance while suppressing astrocyte-mediated neuroinflammation may help mitigate the disease progress of PD.

Notably, several proteins whose mutations are associated with PD, e.g., PTEN-induced putative kinase 1 (PINK1), parkin RBR E3 ubiquitin protein ligase (PRKN), leucine-rich repeat kinase 2 (LRRK2), and glucosylceramidase beta 1 (GBA1), are highly expressed in astrocytes, suggesting the potential dysfunction of astrocytes in PD pathology. Indeed, those proteins exert key functions in important cellular processes such as autophagy, mitochondrial homeostasis, and inflammatory responses. For instance, PD patient-derived astrocytes exhibited abnormal metabolism, e.g., increased levels of polyamines and decreased levels of lysophosphatidylethanolamine [137,138,139]. Also, patient-derived astrocytes with LRRK2 or GBA1 mutations would become more prone to inflammatory responses [140,141].

#### 4.3.3. Amyotrophic Lateral Sclerosis (ALS)

Reactive astrocytes are commonly observed in neural tissues of ALS patients [142,143,144]. Accordingly, astrocytes from ALS patients or mouse models show a dampened ability to clear reactive oxygen species while releasing pro-inflammatory cytokines and chemokines, which can be a key driver of disease by exacerbating the death of motor neurons [145,146,147]. In addition, astrocytes may uptake and then release the misfolded pathological proteins implicated in ALS pathogenesis, e.g., superoxide dismutase 1 (SOD1) and TAR DNA binding protein (TARDBP, also known as TDP-43), thus potentially contributing to disease propagation within the CNS [148,149]. Importantly, normal interactions between astrocytes and motor neurons, which are crucial for maintaining neuronal survival and functions, are disrupted in ALS. Particularly, resting astrocytes are responsible for the timely removal of glutamate from the synaptic cleft through the specific transporters solute carrier family 1 member 2 (SLC1A2, also known as EAAT2) and member 3 (SLC1A3, also known as EAAT1) [43,150]. However, the expression and function of these transporters in astrocytes are impaired during ALS pathology, leading to excess extracellular glutamate, which causes aberrant synaptic transmission and excitotoxic death of motor neurons [151,152,153].

Unlike the scenario of AD or PD, astrocytes may even be a primary trigger of ALS pathology. Astrocytes carrying ALS-causing mutations, such as SOD1, fused in sarcoma (FUS), or C9orf72, would exhibit profound in vitro toxicity towards co-cultured motor neurons, impairing their neurite outgrowth, synaptic integrity, and overall survival [154,155,156,157,158]. In addition, astrocyte-specific deletion of disease-related SOD1 mutant protein is sufficient to ameliorate the loss of motor neurons in mouse models [159]. Therefore, astrocytes and their inflammatory responses associated with those mutations may have a determinant role in the pathogenesis and progression of ALS.

#### 4.3.4. Other Neurological Conditions

Reactive astrocytes occur in diverse neurodegenerative disorders, e.g., Huntington’s disease [160,161], ataxia [162,163], spinal muscular atrophy [164,165], and glaucoma [166,167]. Also, reactive astrocytes can be observed in common autoimmune disorders in the CNS, e.g., autoimmune encephalitis [168,169] and myelin oligodendrocyte glycoprotein antibody-associated disease (MOGAD) [170,171]. While those neurological conditions significantly differ regarding inflicted regions and mechanisms, astrocyte-mediated neuroinflammation and its potential impact on neuronal survival and functions may share similar features as those encountered in the disease conditions described above.

## 5. Future Perspectives

In the past decades, we have witnessed significant advances in the knowledge of astrocyte-mediated neuroinflammation in neurological diseases. It has become evident that astrocytes have complex crosstalk with neurons, oligodendrocytes, microglia, and other cell types in the CNS. This network of intercellular communications designates the overall outcome of astrocyte-mediated neuroinflammation. However, several critical questions still await being addressed by future research.

### 5.1. Translational Values of Animal Models

Current animal models of neurological diseases mostly rely on rodents. In particular, the development of astrocyte-specific reporter lines, e.g., enhanced green fluorescent protein (EGFP) or Cre recombinase driven by the promoter of aldehyde dehydrogenase 1 family member L1 (ALDH1L1), has facilitated the investigations of pathophysiological functions of astrocytes [172]. However, it has become recognized that human and rodent astrocytes exhibit many differences that may have profound implications for translational research. For instance, human astrocytes show greater susceptibility to oxidative stress than their mouse counterpart due to different detoxification pathways [173]. This increased vulnerability of human astrocytes may contribute to more severe neurological outcomes observed in human patients compared to mouse models. Also, the same study revealed that inflammatory stimuli can elevate the expression of antigen presentation-related genes in human but not mouse astrocytes [173]. This species-specific difference in the functional link of innate and adaptive immune responses may complicate the translation of astrocyte-mediated neuroinflammation from mouse models to patients. Therefore, a comprehensive comparison of human and rodent astrocytes on epigenomic, transcriptomic, and proteomic levels will provide more confidence in the animal models of astrocyte biology. In addition, the refinement of chimeric rodent models with human astrocytes may represent a robust alternative solution to this issue [174,175,176].

### 5.2. In Vitro Models of Astrocyte Biology

Studies on human astrocytes have commonly utilized the in vitro cultures of primary astrocytes from healthy donors or astrocytes derived from the induced pluripotent stem cells (iPSCs) of patients. Despite the indispensable value of such cell culture models, it is essential to realize that, as discussed above, the pathophysiological functions of astrocytes often depend on specific tissue microenvironments. Future research exploiting the organoid cultures that combine a variety of cell types, e.g., astrocytes, microglia, vascular endothelial cells, and neurons, may better recapitulate the in vivo complexity of neurological conditions encountered in human patients [177,178,179].

### 5.3. Diagnostics of Astrocyte-Mediated Neuroinflammation

While astrocytes have been well documented to release a repertoire of cytokines and chemokines under neurological insults, the detection of those immune factors in human patients remains difficult. Current approaches primarily rely on sampling the cerebrospinal fluid, which cannot distinguish the astrocytic origin from other cellular sources, e.g., microglia or infiltrating immune cells. The development of specific radiolabeled probes for the positron emission tomography (PET) imaging of astrocyte-mediated inflammatory processes will significantly expand our understanding of their pathophysiological functions in CNS diseases [180,181,182,183]. In addition, high-throughput transcriptomic or proteomic analyses of molecular changes in the astrocytes derived from patient samples or mouse models will provide the comprehensive characterization of astrocyte-mediated neuroinflammation at unparalleled spatiotemporal resolutions.

### 5.4. Therapeutic Strategies for Targeting Astrocytes

It has been postulated that enhancing the neuroprotective functions of astrocytes while reducing their neurotoxic effects may represent powerful strategies against neurological diseases [184,185]. However, it is practically challenging to target astrocytes in human patients, and manipulating astrocyte-specific events for therapeutic benefits still awaits future efforts. The recent development of the recombinant adeno-associated virus serotype 9 (AAV9) or its derivatives, which can effectively transduce astrocytes, may emerge as a robust approach [186,187]. Also, antibody-based drug delivery may offer an alternative avenue to targeting reactive astrocytes in neurological conditions [188,189], e.g., an antibody that specifically recognizes a cell-surface marker unique to those astrocytes can be utilized to deliver small-molecule conjugates. In addition, potential astrocyte-based cell therapies as novel treatments against neurological diseases represent a promising future direction [190,191].

### 5.5. Astrocytes in Psychiatric Disorders

Although astrocyte-mediated neuroinflammation in neurological conditions has been extensively investigated, whether those glial cells may be involved in psychiatric disorders remains incompletely understood. Emerging evidence has implicated the dysregulation of astrocytes in common psychiatric disorders, e.g., depression, anxiety, schizophrenia, and addictive disorders [192,193,194]. For instance, post-mortem studies on the brain tissues of patients with major depressive disorder revealed widespread changes in the morphology of astrocytes and their decreased density, though potentially affected subpopulations require more detailed examinations [195]. Similarly, altered morphology and functions of astrocytes could be observed in several mouse models of depressive-like behaviors [196,197,198]. In addition, antidepressants such as ketamine may exert their therapeutic effects at least partially by targeting astrocytes in such mouse models [199,200,201]. It thus appears a tempting possibility that neuroinflammatory responses of astrocytes may contribute to the onset and progression of those psychiatric disorders, which warrants more research attention.

In sum, this review has summarized the key features of astrocyte-mediated neuroinflammation and highlighted critical questions that call for future research. The combination of enduring basic research and translational efforts on this frontier topic will help reveal novel diagnostic or therapeutic strategies against those debilitating CNS diseases.

## Figures and Tables

**Figure 1 biomolecules-14-01204-f001:**
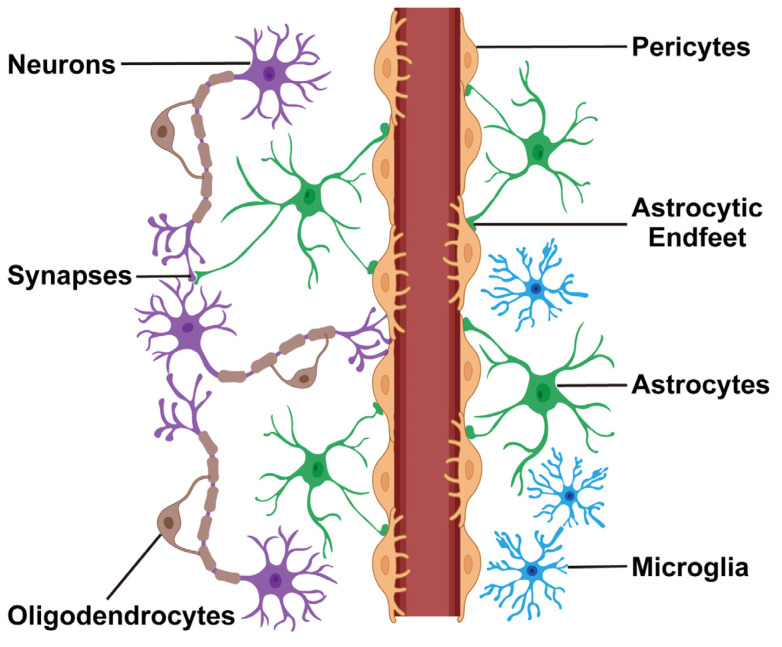
Physiological functions of astrocytes in the CNS. Resting “homeostatic” astrocytes extend their processes to ensheath blood vessels, forming endfeet that maintain the structural integrity of the blood–brain barrier and regulate its function through the release of astrocyte-derived factors. In addition, astrocytes have perisynaptic processes that directly contact neurons and their synapses, which provide critical trophic support and modulate synaptic transmissions.

**Figure 2 biomolecules-14-01204-f002:**
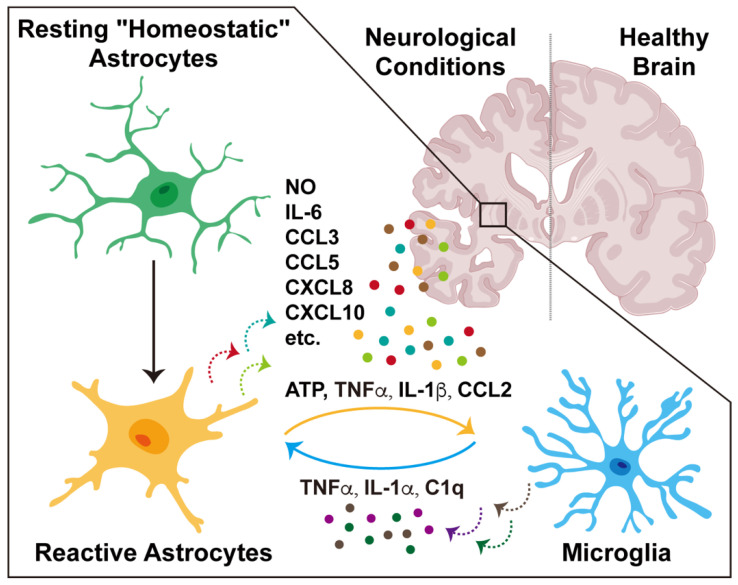
Astrocyte-mediated neuroinflammation in neurological conditions. Resting “homeostatic” astrocytes become reactive astrocytes under neurological conditions, undergoing morphological changes and upregulation of inflammatory cytokines, chemokines, and other signaling factors. In addition, astrocyte-mediated neuroinflammation can engage in bidirectional crosstalk with microglia in the CNS microenvironment. Details are described in the main text.

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
