# Peer review of "Astrocyte-Mediated Neuroinflammation in Neurological Conditions"

_biomolecules, 2024, doi:10.3390/biom14101204_

Round 1

Reviewer 1 Report

Comments and Suggestions for Authors
  • A brief summary 

This manuscript comprehensively reviews the role of astrocytes in neuroinflammation in several neurological conditions and points out several aspects for future astrocyte-related research.

  • General concept comments
    The review cited recent publications and was well structured. However, two similar reviews with much more detailed pathways were published. More summaries on the astrocyte-microglia and astrocyte-neuron crosstalk will differentiate this review from others.
    https://www.nature.com/articles/s41392-023-01628-9
    https://www.nature.com/articles/s41573-022-00390-x
  • Specific comments 
    1.
    Section 3, line 118: GFAP is a reactive astrocyte marker, so are there any resting astrocyte markers?

2. Section 4.3.1: This study (10.1038/s41593-020-0624-8) not only identified a new AD-related astrocyte subtype but also highlighted its significance, making its inclusion in this review crucial.

3. Section 5.1: Adding the discussion on the animal model with astrocyte-specific Cre to drive the knockout genes of interest in the astrocytes also attracts readers who want to study astrocyte-related genes, such as the Aldh1l1-Cre/ERT2 line.

4. Section 5.4, line 352: Add the accurate number of the new AVV serotypes that target astrocytes.

5. Line 365: Is it possible to categorize the morphology of different astrocytes and the intensity of which astrocyte subtype was decreased?

Reviewer 2 Report

Comments and Suggestions for Authors

The manuscript by Zhao et al. reviews astrocyte function in physiological and neurological conditions. Astrocytes are abundant in the human CNS, and their role in maintaining CNS homeostasis is indispensable. Their immunomodulatory role is emerging, and it is of high interest to address this topic.

The "Future perspectives" section and mentioning the intraspecies differences in astrocyte functions are positive aspects of the manuscript.

The largest part of the cited literature is recent.

However, there are several major and minor issues that could be addressed to improve this review article:

1) In the abstract, the authors mention that astrocytes are one of the major glial types. How do they define "major" and "minor" glial cell types?

2) The cross-talk of astrocytes with neurons and microglia is discussed, but there is a lack of discussion on their interaction with oligodendrocytes. This is important in the context of de- and remyelination.

3) The review predominantly focuses on the effects of astrocyte-produced neuroinflammatory molecules and their impact, but it would be beneficial to have a better description of the activation of astrocytes in neuroinflammatory conditions. The triggers are briefly mentioned, but they should be elaborated on in more detail.

4) Given the focus of this review on astrocyte reactivity in neuroinflammation and degeneration, it would be good to include multiple sclerosis and Huntington's disease in the list of described diseases.

5) While astrocytes as therapeutic targets are briefly mentioned, their therapeutic perspective should be addressed in greater detail. Both in the context of existing therapies and especially in the context of new technical advances and findings, such as high-throughput screening of changes in astrocyte function and synthetic engineering. Furthermore, the potential cell therapies using astrocytes that are primed to be neuroprotective should be mentioned.

6) The review should also address region-specific functional heterogeneity.

7) Line 357 - a citation needs to be included.

8) In the text, much is described about the reaction of astrocytes in the pathological environment (secretion of soluble immune mediators). However, more information about the molecular bases and intracellular pathways of their harmful or beneficial function should be stated.

The literature about functional heterogeneity of astrocytes is emerging and should be better described in this review. It has been only briefly mentioned.

Is something known about the roles of other cell types in the initiation, potentiation and cessation of reactive astrocyte phenotypes?

9) Information on mutations in human astrocytes known to contribute to pathology in the context of described diseases should be included.

Comments on the Quality of English Language

The language used throughout the manuscript is adequate.

Round 2

Reviewer 2 Report

Comments and Suggestions for Authors

Thank you for addressing my comments.

Comments on the Quality of English Language

Minor editing of English language required.